# Acute Pesticide Poisoning in Tobacco Farming, According to Different Criteria

**DOI:** 10.3390/ijerph20042818

**Published:** 2023-02-05

**Authors:** Neice Muller Xavier Faria, Rodrigo Dalke Meucci, Nadia Spada Fiori, Maria Laura Vidal Carret, Carlos Augusto Mello-da-Silva, Anaclaudia Gastal Fassa

**Affiliations:** 1Department of Social Medicine, Faculty of Medicine, Federal University of Pelotas, Pelotas 96030-000, Brazil; 2Faculty of Medicine, Federal University of Rio Grande, Rio Grande 96203-900, Brazil; 3Poison Information Center (CIT/RS), State Health Department, Porto Alegre 90610-000, Brazil

**Keywords:** pesticide, poisoning, epidemiology, toxicology, tobacco, farmer

## Abstract

Background: Brazil is one of the world’s largest pesticide consumers, but information on pesticide poisoning among workers is scarce. Objective: To evaluate acute pesticide poisoning among tobacco growers, according to different criteria. Methods: This was a two-step cross-sectional study with 492 pesticide applicators. It used a 25 question pesticide-related symptoms (PRS) questionnaire and medical diagnosis for comparison with toxicological assessment. Associations were evaluated using Poisson regression. Results: 10.6% reported two or more PRS, while 8.1% reported three or more. Furthermore, 12.2% received a medical diagnosis of poisoning. According to toxicologists, possible cases accounted for 14.2% and probable cases for 4.3%. PRS increased during the period of greater exposure. Those exposed to dithiocarbamates, sulfentrazone, pyrethroids, fipronil and iprodione exhibited more PRS. The number of exposure types, multi-chemical exposure, clothes wet with pesticides and spillage on the body/clothes were associated with acute poisonings. All criteria showed sensitivity greater than 79% for probable cases but only greater than 70% for medical diagnosis when compared to possible cases, presenting substantial Kappa agreement. Conclusion: The prevalence of acute pesticide poisoning is much higher than officially recorded. Trained physicians can screen for pesticide poisoning. It is necessary to improve workers’ education to reduce pesticide use and exposure to them.

## 1. Introduction

For decades now, Brazil has been one of the world’s largest consumers of pesticides [1,2]. Consumption of active ingredients per hectare of planted area increased from 2.9 kg/hectares in 2003 to 6.7 in 2014 [3]. According to the Brazilian Institute of Environment and Renewable Natural Resources (Instituto Brasileiro de Meio Ambiente—IBAMA), in 2021, a total of 720,869 tons of formulated pesticides were sold, which represents a 50% increase compared to 2012 [3]. This highly chemical agricultural production model has considerable impacts on the environment, such as soil and water contamination. Biocides also generate threats to living beings, especially pollinating insects, which contribute directly to agricultural production [4], in addition to negatively affecting birds and mammals, including humans.

Agricultural workers have frequent occupational and environmental exposure, often multi-chemical, and are at increased risk of pesticide poisoning and death. Estimates of acute poisoning prevalence vary depending on the source. According to the Notifiable Diseases Information System (Sistema de Informação de Agravos de Notificação—SINAN), in 2019 in Brazil, 7.7 cases of pesticide poisonings were reported per 100,000 inhabitants, including poisonings from products used in agriculture, public health, household pesticides, rodenticides and veterinary products [5].

Epidemiological studies indicate a much higher occurrence of pesticide poisoning among agricultural workers. Studies conducted in the Serra Gaúcha region with family farmers identified 2.0 to 3.8% who reported acute pesticide poisoning in the year before they were interviewed [6,7]. In addition, several studies have pointed to association between acute poisoning episodes and increased prevalence of chronic diseases, such as respiratory problems [8,9,10,11], hormonal disorders [12], neurological disorders [13] and mental disorders [14,15,16,17,18,19].

Records of pesticide poisoning are limited by difficulties in access to health services, especially in rural areas; failure to recognize cases and low adherence to case notification. Estimates of occurrence of pesticide poisoning in epidemiological studies are affected by the diversity of criteria used to identify and classify acute poisoning, with self-reported poisoning being the most frequently used. With the aim of increasing the comparability of findings, the World Health Organization (WHO) has proposed a methodology to classify acute pesticide poisonings as either possible cases or probable cases [20].

This study evaluates prevalence of acute pesticide poisoning among family farmers who applied pesticides, using a pesticide-related symptoms (PRS) questionnaire (developed according to the methodology proposed by the WHO), clinical physician diagnosis and toxicological assessment. The study also investigates occupational factors associated with acute pesticide poisoning.

## 2. Methods

### 2.1. Study Design and Participants

A cross-sectional study was carried out in São Lourenço do Sul, Rio Grande do Sul, the state that most produces tobacco in Brazil [21]. The study had three stages. The sample procedures took into account the sample size needed for the third stage, which addressed green tobacco sickness during harvest time and required the largest sample size. For sampling, 1100 of the 3852 tobacco sales invoices issued in 2009 were randomly selected. This article presents a subsample analysis of family tobacco growers aged 18 years or over, who applied pesticides, participated in the first and second stage of the study and worked in the 254 sampled farms in the Canta Galo and Santa Inês districts, which are the largest tobacco producers in the municipality of São Lourenço do Sul. The first stage occurred in the period of low exposure to pesticides, the second in the period of high exposure

The vast majority of farms (82%) were up to 40 hectares in size. The production context of the areas studied was detailed in previous publications [15,21,22]

### 2.2. Sociodemographic and Occupational Aspects

We investigated demographic aspects, i.e., gender and age; socioeconomic aspects, i.e., education in complete years of study and annual tobacco production in tons; behavioral aspects, i.e., smoking (never smoked, ex-smoker and smoker), alcohol consumption (in days/month) and Body mass index (BMI) (normal weight, overweight and obesity) [20].

### 2.3. Pesticide Exposure

Occupational exposure to pesticides in the month prior to the interview was characterized in number of days. Several forms of exposure were investigated in relation to the most recent contact with pesticides, such as applying herbicides or insecticides/fungicides, mixing/preparing mixture, cleaning equipment, washing contaminated clothing, having contact in transportation and storage, through wet clothing, reentering into the plantation without protection after application, and spilling on the body or on clothing. We also characterized the sum of the forms of pesticide exposure.

In order to gather information on what types of chemicals were used on the farms in the last 30 days, plastic cards were prepared with pictures of the main pesticides used in tobacco farming in the region, totaling 56 commercial formulations and more than 22 products (referred to as others). The date of most recent contact was also collected for each product reported. The chemical groups analyzed were organophosphorus compounds (organophosphates), neonicotinoids, pyrethroids, carbamates, benzoylurea, fipronil, clomazone, glyphosate, sulfentrazone, triazines, dithiocarbamates, metalaxyl, iprodione and copper compounds. The sum of the types of chemicals used was taken as an indicator of multi-chemical exposure.

The personal protective equipment (PPE) investigated were boots, masks and gloves to protect against chemicals, protective clothing, hat/headgear, characterizing the number of days they were used during work in the last 30 days. We also examined the sum of the PPE that each worker reported always using.

### 2.4. Pesticide Poisonings

In both stages, 25 symptoms common to pesticide poisonings that occurred in the last 15 days were investigated. Pesticide-related symptoms (PRS) were considered when they had appeared or worsened within 48 h after direct contact with these products. According to the classification proposed by the World Health Organization (WHO), a possible case of poisoning was considered to be when two or more PRS were present, while a probable case was considered to be when three or more PRS were present [20]. Medical assessment included standardized anamnesis and physical examination and was performed by clinical physicians trained by the study. The standardization of the medical evaluation instruments and the criteria to define whether or not the worker had acute pesticide poisoning were developed in a workshop on the subject with the participation of toxicologists and epidemiologists.

The toxicological assessment was performed by two toxicologists, and in case of disagreement, a third toxicologist also gave an opinion. The toxicological assessment took into consideration the PRS questionnaire, the medical assessment, and a set of complementary tests. Anthropometric data, vital signs, Gamma GT (GGT), and casual capillary blood glucose were collected. Plasma cholinesterase (Butyrylcholinesterase—BCHE) samples were also collected, whereby the measurement taken in the first stage was considered to be baseline BCHE, while the measurement taken in the second stage was considered to be measurement at high exposure. In cases of recent exposure to cholinesterase inhibitors, the BCHE measurement was considered in the toxicologists’ diagnosis. In cases with decreased BCHE (at least a 20% reduction in relation to baseline BCHE) and/or altered GGT (≥60 U/L), abdominal ultrasonography, liver function tests, and viral hepatitis serology were performed.

The toxicologists considered a probable case of poisoning to be when data were compatible with the expected clinical manifestations, considering the pesticides to which the tobacco grower was exposed in the period. They considered a possible case of poisoning to be when the data could be compatible with the expected clinical manifestations, considering the pesticides to which the tobacco grower was exposed in the period, but the available data did not allow other causes to be excluded (uncertainty about differential diagnoses).

### 2.5. Statistic Analysis

The data analysis described the study population, indicating the prevalence of pesticide poisoning in each population subgroup according to the different criteria studied. It then compared prevalence of PRS in the first and second stages, estimating the relative risk and confidence interval for the occurrence of the symptom in the second stage (high pesticide exposure) compared to the first stage (low exposure). It also examined whether there was a significant difference in the prevalence of symptoms among those exposed to each type of chemical evaluated when compared to those not exposed, using Pearson’s chi-square test or Fisher’s exact test for categories with numbers less than five.

Statistical analysis also included examining the factors associated with pesticide poisoning according to the different criteria by examining association using the chi-square test for categorical variables and the linear trend test for continuous variables. The adjusted analysis was performed using Poisson regression with robust variance. We used a hierarchical model containing demographic and socioeconomic aspects in the first level and behavioral aspects and occupational exposures to pesticides in the second level, keeping in the model variables with p-value less than or equal to 0.2. To avoid overlapping effects, occupational exposures to pesticides were examined one at a time, adjusting for the first level and behavioral variables.

Analysis of sensitivity, specificity, positive and negative predictive value was performed, taking the toxicological assessment as the gold standard. Agreement between the different diagnostic criteria in relation to the toxicologists’ assessment as to possible case and probable case was examined using the Kappa test.

### 2.6. Ethical Issues

This study was approved by the Research Ethics Committee of the Medical School of the Federal University of Pelotas (Universidade Federal de Pelotas—UFPEL) as per letter number 11/2010. Eligible farm workers were informed about the research topic, the right to not take part, and the guarantee of confidentiality regarding individual information. Those who agreed to participate signed an informed consent form. Interviewees who were identified as having acute pesticide poisoning were referred for treatment in the public health system.

## 3. Results

### 3.1. Participants

A total of 492 tobacco growers who had applied pesticides in the previous year took part in the study. In the first stage, losses and refusals accounted for 5% in relation to those who were eligible, while in the second stage, they accounted for 4% in relation to the first stage. Ninety-eight percent were white, most were of German descent, 90% of whom belonged to the farm owner’s family and the rest were tenants. Almost all (94.3%) had contact with pesticides in the last 20 days (data not shown).

### 3.2. Sociodemographic and Occupational Aspects

Among the tobacco growers, 26% were under 30 years old and 25.6% were 50 years old or over, 76.4% were male, 44% had 4 complete years or less of schooling, 29.5% consumed alcoholic beverages 8 days or more per month, 19.7% were smokers, and 12.6% were overweight. Among the farmers, 70.1% produced up to 9000 kg of tobacco per year. Regarding forms of exposure to pesticides, 74.7% prepared mixture, some 60% applied pesticide and/or cleaned equipment, 52.8% reported reentering into the plantation after pesticide application without using PPE, 45.3% washed contaminated clothes, 33.3% reported that their clothes got wet with pesticide while applying it, and 20.4% reported spilling pesticides on their bodies or clothes. Among the tobacco growers, 40.8% had 7 to 12 forms of exposure to pesticides, 32.5% were exposed to 5 or more types of chemicals, 24.6% were exposed for 5 days or more per month, and 37.2% always used four or five PPE items (Table 1).

### 3.3. Acute Pesticide Poisonings Criteria

Among the tobacco growers,10.6% reported two or more PRS, 8.1% reported three or more PRS, 12.2% received a medical diagnosis of acute pesticide poisoning. According to the toxicologists’ assessment, 14.2% were classified as possible cases and 4.3% as probable cases of pesticide poisoning (Table 1). The prevalence of self-reported pesticide poisoning in the last year was 2% (data not shown).

### 3.4. Pesticide Related Symptoms—PRS

The symptoms that were significantly more frequent in the second stage when compared to the first stage were loss of appetite (RR/risk ratio 3.60); headache (RR 3.50); eye irritation (RR 3.29); salivation (RR 3.19); dyspepsia/difficult digestion (RR 2.82); blurred vision (RR 2.71); skin allergy (RR 2.38) and dizziness (RR 2.10) (Table 2).

### 3.5. Pesticide Chemical Types and PRS

The types of chemicals that farmer workers reported using the most with regard to insecticides were neonicotinoids (53.7%), organophosphates (48.6%) and pyrethroids (14,8%); with regard to weedkillers, they used clomazone (56.5%), glyphosate (30.5%) and sulfentrazone (25.6%); while with regard to fungicides, they used dithiocarbamates (38.2%), metalaxyl (30.3%) and iprodione (20.3%). Exposure to dithiocarbamates was significantly associated with an increase in nine PRS, exposure to sulfentrazone with an increase in eight PRS, exposure to pyrethroids with an increase in six PRS, exposure to iprodione and copper compounds with an increase in three PRS, and exposure to fipronil with an increase in two PRS (Table 3).

### 3.6. Bivariate Analysis

In the bivariate analysis, women presented with 2.5 times more pesticide poisonings than men according to the PRS criterion and the toxicologists’ assessment as probable cases. Among those whose tobacco production was up to 4000 kg, the frequency of the outcome was 2.5 times higher than those who produced more than 9000 kg according to the criteria of two or more PRS, medical diagnosis and probable case as per the toxicologists’ assessment. Alcohol consumption was protective for poisoning, except for the criterion of possible case as per the toxicologists’ assessment (Table 4). Age, schooling, BMI and smoking showed no association with any of the criteria.

As for the forms of exposure, having clothes wet with pesticide during application and suffering pesticide spillage on the body or clothes were associated with the outcome, with risks higher than 68% according to all criteria, reaching a risk of 300% for clothes wet with pesticide and 500% for spillage on the body or clothes according to the probable case criterion. Applying pesticides and going back into the plantation without PPE were associated with an almost two-fold increase in pesticide poisoning according to the two or more PRS and three or more PRS criteria. Washing contaminated clothing was associated with the outcome with risks ranging from 76% to 200% using the different criteria, except for the possible case criterion, for which there was no association (Table 4).

The number of forms of worker exposure showed significant direct association with the outcome (linear trend p-value less than 0.01), except for the probable diagnosis criterion, which did not have statistical power to perform the calculation. The same occurred with the number of types of chemicals used, except for the medical diagnosis criterion. The number of days of pesticide use showed direct association and the number of PPEs always used showed inverse association with the outcome according to the two or more PRS and three or more PRS criteria (Table 4).

### 3.7. Multivariate Analysis—Poisson Regression

The multivariate analysis was adjusted for gender, tobacco production and alcohol consumption. Alcohol consumption and washing contaminated clothes lost their association with the outcome, while being female and the number of days of pesticide use also became associated according to the possible case criterion as per the toxicologists’ assessment. Pesticide poisoning was almost double among those exposed to pesticide solution preparation according to the two or more PRS and possible case criteria. Multi-chemical exposure not only remained associated with the PRS and toxicological criteria but also showed increased strength of association in those exposed to five or more types of pesticides. The other associations remained similar to those of the bivariate analysis (Table 5).

### 3.8. Sensitivity, Specificity, Predictive Value and Agreement

Table 6 shows the analysis of sensitivity, specificity, predictive value and agreement for two or more PRS, three or more PRS and medical diagnosis when comparing possible cases and probable cases of pesticide poisoning as per the toxicologists’ assessment. All diagnostic criteria exhibited specificity above 95% for possible cases and above 90% for probable cases. The negative predictive value was greater than 90% for possible cases and 99% for probable cases. In the comparison with probable cases, the criteria that used PRS exhibited less than 50% sensitivity and a positive predictive value less than 61%, while medical diagnosis showed 73% sensitivity and an 85% positive predictive value. With regard to probable cases, sensitivity was greater than 79% for all criteria and positive predictive value was less than 39% for all criteria. Agreement as evaluated using the Kappa test was substantial for the comparison between medical diagnosis and possible cases (0.75) and moderate for three or more PRS compared with probable cases (0.49) and for medical diagnosis compared with probable cases (0.43).

## 4. Discussion

The study indicates that the different criteria used result in variability in the estimates of acute pesticide poisoning, all of which are higher than self-reported information, which in turn, are higher than the official records that identified, at the most, 11 annual cases per 100,000 inhabitants [5,23]. In the Serra Gaúcha fruit growing, the prevalence of acute pesticide poisoning also varied according to the criterion used, being 3.8% in 12 months according to workers’ self-reported information and 11% for probable cases according to the criterion of three or more PRS [7]. In the state of Espírito Santo, 7.5% of agricultural workers exhibited poisoning, based on self-reported previous medical diagnosis [19].

### 4.1. Pesticide-Related Symptoms-PRS

Questionnaires of PRS were used as a criterion for identifying pesticide poisoning in other studies, although with some differences in the list of symptoms [7,24,25,26,27,28] and in the criterion regarding the time between exposure and occurrence of symptoms, which varied from 24 h [25,26] to 48 h [24,28], or recent symptoms without specifying the period of time following pesticide use [7,27], or even presence of common poisoning symptoms without considering the time relationship with exposure [29]. Although all the PRS showed some degree of increase in the second stage (greater exposure to chemicals), the increase was only significant for some of the symptoms, possibly due to statistical power limitations. The symptoms assessed are common to several morbidities, but the exposure time criterion, according to the WHO methodology, suggests a link with occupational exposure, with some effects being caused by contact with chemicals, such as eye and skin symptoms, and others being systemic, such as neurological and digestive symptoms.

### 4.2. Pesticide Chemical Types

Among the types of chemicals that stood out in the associations with PRS, most are considered to have low acute toxicity, such as dithiocarbamates, sulfentrazone and iprodione, for example [30,31]. Most of these chemicals can cause skin and mucous membrane (respiratory, ocular and digestive) symptoms. Dithiocarbamate fungicides (especially mancozeb) are associated with acute poisoning; skin, respiratory system and ocular mucous membranes irritation, and with systemic effects, such as thyroid alterations, liver disease and kidney disease [32]. The weedkiller sulfentrazone, a lesser known pesticide, in addition to skin and mucous membrane symptoms, is linked to developmental and reproductive effects [31]. With regard to iprodione, studies with animals indicate that it has a carcinogenic effect and reproductive toxicity [33].

Pyrethroids are widely used insecticides, including in urban areas, and their acute allergic, respiratory and skin effects are well known, as well as systemic effects such as oxidative stress and neurotoxic, hepatotoxic and nephrotoxic symptoms, among others [34]. Fipronil is another increasingly used insecticide and can even be found in pet shops [35]. This insecticide is a GABA inhibitor (gamma-aminobutyric acid), which can produce neurological (headache, dizziness, paresthesia), ocular, gastrointestinal, respiratory and skin symptoms [36].

Occurrence of PRS can be due to either the individual effect or the combined effect of the various types of chemicals, as well as other components of the commercial formulation, such as surfactants and solvents. As such, multi-chemical exposure makes it difficult to identify the specific pesticide related to each symptom. It is possible that some associations (or the lack of them) may have been influenced by frequent exposure, by multi-chemical exposure and/or by limitations in the measurement of exposure. Toxicological research generally tests one product at a time and considers primarily the risk of death, assigning less toxicity to other effects. In addition to the paucity of information on the toxic effects of some types of chemicals on humans, there are fewer studies on the mild and moderate effects typical in the occupational context. The workers who exhibited PRS were active and displayed mild conditions with no need for referral to hospital or emergency services. For this reason, they may have shown some symptoms that are not among the main ones identified in severe cases.

### 4.3. Acute Pesticide Poisoning—Main Associated Factors

The higher prevalence of acute pesticide poisoning among women is consistent with other studies that have examined PRS among agricultural workers [7,25,26,37,38] but contrasts with official records in Brazil [5,23,39,40,41], in which pesticide poisonings are predominant among men. Official records may underestimate more the occurrence of poisoning among women, due to greater failure to investigate their occupation and/or exposure to pesticides, or by not considering common forms of female exposure, such as washing contaminated clothes. Hormonal mechanisms, smaller body area, lower adherence to PPE, and the fact that women report their health problems better may explain higher prevalence of poisoning among women [15]. Considering that pesticide application is a predominantly male task, this study assessed a particularly exposed subgroup of women, given that they had applied pesticides in the last year.

Inverse association between tobacco production and pesticide poisoning is consistent with the literature, which indicates that poorer socioeconomic conditions are related to poorer health conditions [42,43]. Farms with better economic indicators usually have a better level of technology, including pesticide application equipment, more adequate personal protection equipment, and better agricultural practices, implying a reduction in exposure. As it occurred in China [25], in our study there was no association between schooling and poisoning. This contrasted with previous studies in the Serra Gaúcha region that pointed to an inverse association between schooling and acute poisoning [6,7], as well as with other studies in which low schooling was associated with higher exposure to pesticides and unsafe practices during work involving pesticides [44]. There was limited statistical power to assess this association, especially since the vast majority (86.8%) of the workers studied had not completed elementary school.

In occupational exposure, the skin is the most important pesticide absorption route [28,45,46], and in hot climates, such as in tropical countries, absorption is even greater [25]. Several pesticides, such as organophosphorus compounds, carbamates and pyrethroids, demonstrate considerable cutaneous absorption [20,34,47,48]. In keeping with other studies [46,48,49], spilling pesticide on the body and wearing clothes wet from pesticides were associated with all criteria with great strength of association, especially the toxicological probable case criterion. Other studies indicated that poisoning was greater among workers who went back into the plantation without protection following spraying [28,49]. In our study, this aspect was associated with PRS. The direct association between number of chemical types and number of forms of exposure to pesticides with regard to all the poisoning criteria suggests that these may be indicators of exposure intensity.

### 4.4. Personal Protective Equipment—PPE

Some studies, especially clinical trials [45] focusing on PPE testing, point to their protective effect, while others question it [50,51]. Furthermore, some population-based studies [7] have not confirmed their protective effect, suggesting that it may be overestimated [52]. The inverse association between PPE use, found only with the PRS-based criteria and no significant association with the other criteria, led us to question the effectiveness of PPE. On the other hand, the lack of information concerning protection of the eyes, face and neck; the constancy, the adequacy and completeness of PPE used for specific forms of exposures; as well as the lack of information about PPE quality, PPE cleaning practices and PPE use in the different forms of exposure, limit evaluation of PPE. Although tobacco farmers report receiving frequent technical guidance, they do not always use PPE as recommended. Discomfort when using PPE in high temperatures, as well as its cost, are major factors for low adherence to PPE use [53].

### 4.5. Pesticide Poisoning Criteria Evaluation

The PRS questionnaire, considering three or more symptoms, has proven useful for probable case screening but does not adequately capture possible cases of pesticide poisoning. On the other hand, medical diagnosis by trained professionals, when compared to the toxicologists’ assessment, has adequate sensitivity and agreement for screening both probable and possible cases, even without laboratory tests. It should be noted that the toxicologists’ assessment, the other standard used, took into consideration all the other criteria and also presents a certain degree of subjectivity.

### 4.6. Limitations and Challenges

This study contained a representative sample of tobacco growers who applied pesticides, and the sample size was adequate for assessing prevalence of acute pesticide poisoning according to different criteria. However, there were limitations in the statistical power of the study to examine some associations, especially with the probable case outcome and in the comparison between the two stages. In addition, since all of them had applied pesticides in the previous year, there was little variability in some forms of exposure. Flumethralin (plant growth inhibitor), one of the most used chemical types, was not assessed because the period in which it is used had not started when the study fieldwork took place. It is also noteworthy that the forms of exposure were related to most recent contact with pesticides, which may have reduced the proportion of tobacco growers who reported doing certain tasks, such as washing contaminated clothes. The self-reported information about symptoms and pesticide exposure is valuable and easy to collect but can be imprecise and subjected to recall bias, in general, underestimating its prevalence. Since pesticide use has increased in Brazil, and given that there has not been much progress in investigating acute pesticide poisoning in the country, despite the study being conducted some time ago, it nevertheless presents relevant and conservative information in relation to the current situation [20].

The availability and cost of biomarkers in Brazil make their inclusion in epidemiological studies difficult, so our study only relied on BCHE testing. The use of cards with pictures of the most common pesticides reduced memory bias and, together with the detailing of exposure, ensured information quality. However, an expansion and improvement of laboratories for toxicological tests is needed to qualify clinical and epidemiological evaluations. The partnership with the Municipal Health Department and with Family Health Strategy rural area health teams facilitated the logistics of the study and adherence of the tobacco farmers to the research. In return, the research provided training for health professionals in rural areas about acute pesticide and nicotine poisoning.

In order to obtain greater heterogeneity with regard to exposure, it is important for future studies to include all agricultural workers and not only those who applied pesticides in the previous year. This being a low-prevalence event, it is also essential to have a larger sample size, to enable more in-depth understanding of factors associated with probable cases. It is necessary to move forward with objective characterization of exposure by expanding the use of biomarkers. However, in the absence of biomarkers, it is important to prepare cards with pictures of the most commonly used pesticides in order to characterize their use. Moreover, when evaluating intense exposure, multi-chemical exposure and the number of forms of exposure can be used as indicators. The PRS questionnaire can be used in epidemiological studies to estimate probable cases. During the interviews, we noted that restlessness/irritability symptoms, although they are neurological symptoms, may indicate different neuropsychiatric problems and, therefore, should have been collected separately. For future studies we suggest replacing restlessness/irritability with restlessness/uneasiness and nervousness/irritability. PPE evaluation requires specific studies that allow a better understanding of the factors that reduce its protective role.

## 5. Conclusions and Recommendations

This study indicates an acute pesticide poisoning prevalence much higher than that officially recorded. Considering the high prevalence of poisoning, as well as the high and increasing use of these chemicals, public policies that seek to reduce exposure to pesticides should be encouraged, including the quest for sustainable agricultural production models [29], their replacement with less toxic products, as well as individual protection measures.

Among the main occupational associated factors, those with skin exposure stood out. Thus, special attention should be paid to reduce skin exposure through the implementation of occupational hygiene measures such as the quick removal of wet contaminated clothes, clean hands and skin with pesticide residues, caution when handling pesticides or when removing PPE and correct use of PPE. These approaches should go hand in hand with actions to educate workers about the toxicity of products, reading labels, product handling, forms of chemical exposure and the adoption of good agricultural practices.

This study indicates that the PRS questionnaire is an important tool to medical diagnosis and that trained physicians, even without laboratory tests, can screen for possible and probable cases of pesticide poisoning. Thus, these professionals should be trained to perform early detection, notification and clinical management of people who were poisoned. An open access site with information on the association of pesticides with acute poisoning and chronic effects, maintained by reputable public institutions, could be an important support for health professionals and researchers.

In addition, health teams need to be trained to guide agricultural workers and their families in preventing these forms of pesticide poisoning.

## Figures and Tables

**Table 1 ijerph-20-02818-t001:** Acute Pesticide Poisoning according to criteria: Pesticide-Related Symptoms (PRS), medical diagnosis and toxicologists’ assessment.

Variables	N (%)	2 or More PRS	3 or More PRS	Medical Diagnosis	Toxicologist Possible Case	Toxicologist Probable Case
Total sample	492 (100.0)	51 (10.6%)	39 (8.1%)	60 (12.2%)	70 (14.2%)	21 (4.3%)
Sex		▶*p* = 0.001	▶*p* = 0.001	▶*p* = 0.11	▶*p* = 0.05	▶*p* = 0.03
Male	376 (76.4)	30 (8.1)	22 (5.9)	41 (10.9)	47 (12.5)	12 (3.2)
Female	116 (23.6)	21 (19.3)	17 (15.6)	19 (16.5)	23 (19.8)	9 (7.8)
Age Group		▶*p* = 0.83	▶*p* = 0.52	▶*p* = 0.20	▶*p* = 0.42	*p* = 0.74 +
Up to 29 years	128 (26.0)	13 (10.3)	10 (7.9)	10 (7.9)	14 (10.9)	4 (3.1)
30–49 years	238 (48.4)	23 (10.0)	16 (7.0)	34 (14.3)	38 (16.0)	12(5.0)
50 years and over	126 (25.6)	15 (12.1)	13 (10.5)	16 (12.7)	18 (14.3)	5 (4.0)
Schooling		▶*p* = 0.70	▶*p* = 0.46	▶*p* = 0.49	▶*p* = 0.18	*p* = 0.16 +
Up to 4 years	218 (44.3)	23 (10.6)	17 (7.9)	29 (13.4)	30 (13.8)	10 (4.6)
5–7 years	209 (42.5)	23 (11.6)	19 (9.5)	26 (12.4)	35 (16.7)	11 (5.3)
8 years and over	65 (13.2)	5 (7.8)	3 (4.7)	5 (7.8)	5 (7.7)	0 (0.0)
Tobacco production		*p* = 0.01 *	▶*p* = 0.05	*p* = 0.008 *	*p* = 0.08 *	*p* = 0.03 +
Up to 4000 kg	165 (34.1)	22 (13.8)	14 (8.8)	29 (17.8)	29 (17.6)	13 (7.9)
4001–9000 kg	178 (36.8)	23 (13.1)	20 (11.4)	20 (11.2)	25 (14.0)	4 (2.2)
Over 9000 kg	141 (29.1)	6 (4.4)	5 (3.7)	11 (7.8)	15 (10.6)	4 (2.8)
Alcohol consumption (Days/month)		*p* = 0.03 *	*p* = 0.02 *	*p* = 0.03 *	*p* = 0.07 *	*p* = 0.03 +
Does not drink	116 (23.6)	18 (15.8)	15 (13.2)	18 (15.5)	22 (19.0)	10 (8.6)
Up to 7 days/month	231 (47.0)	23 (10.2)	17 (7.6)	32 (14.0)	32 (13.9)	8 (3.5)
≥8 days/month	145 (29.5)	10 (7.1)	7 (5.0)	10 (6.9)	16 (11.0)	3 (2.1)
Tobacco smoking		▶*p* = 0.68	*p* = 0.26 +	▶*p* = 0.30	▶*p* = 0.21	*p* = 0.53 +
No	307 (62.4)	33 (11.1)	28 (9.4)	35 (11.5)	45 (14.7)	15 (4.9)
Former smoker	88 (17.9)	10 (11.8)	7 (8.2)	15 (17.0)	16 (18.2)	4 (4.5)
Smoker	97 (19.7)	8 (8.2)	4 (4.1)	10 (10.3)	9 (9.3)	2 (2.1)
BMI		▶*p* = 0.58	*p* = 0.46+	*p* = 0.14 *	▶*p* = 0.13	*p* = 0.91 +
Normal weight	238 (48.4)	21 (9.1)	16 (7.0)	24 (10.1)	28 (11.8)	10 (4.2)
Overweight	192 (39.0)	23 (12.2)	19 (10.1)	26 (13.7)	35 (18.2)	9 (4.7)
Obese	62 (12.6)	7 (11.5)	4 (6.6)	10 (16.1)	7 (11.3)	2 (3.2)
Applying pesticides		▶*p* = 0.01	▶*p* = 0.03	▶*p* = 0.09	▶*p* = 0.07	▶*p* = 0.07 +
No	196(40.0)	12 (6.3)	9 (4.7)	18 (9.2)	21 (10.7)	4 (2.0)
Yes	294(60.0)	39 (13.5)	30 (10.4)	42 (14.3)	49 (16.7)	17 (5.8)
Preparing mixture		▶*p* = 0.05	▶*p* = 0.16	▶*p* = 0.10	▶*p* = 0.05	*p* = 0.12 +
No	123(25.3)	7 (5.8)	6 (5.0)	10 (8.1)	11 (8.9)	2 (1.6)
Yes	363(74.7)	43 (12.1)	32 (9.0)	50 (13.9)	58 (16.0)	19 (5.2)
Washing contaminated clothing		▶*p* = 0.03	▶*p* = 0.03	▶*p* = 0.02	▶*p* = 0.13	▶*p* = 0.01
No	268(54.7)	21 (8.0)	15 (5.7)	24 (9.0)	32 (11.9)	6 (2.2)
Yes	222(45.3)	30 (14.1)	24 (11.3)	35 (15.8)	37 (16.7)	15 (6.8)
Cleaning pesticide equipment		▶*p* = 0.74	▶*p* = 0.92	▶*p* = 0.98	▶*p* = 0.45	▶*p* = 0.80
No	174(35.4)	19 (11.3)	14 (8.3)	21 (12.2)	22 (12.6)	8 (4.6)
Yes	317(64.6)	32 (10.3)	25 (8.1)	39 (12.3)	48 (15.1)	13 (4.1)
Reentering after application		▶*p* = 0.01	▶*p* = 0.01	▶*p* = 0.22	▶*p* = 0.78	▶*p* = 0.97
No	230(47.2)	16 (7.1)	11 (4.9)	24 (10.4)	32 (13.9)	10 (4.3)
Yes	257(52.8)	35 (14.1)	28 (11.3)	36 (14.1)	38 (14.8)	11 (4.3)
Clothes wet during application		▶*p* = 0.01	▶*p* = 0.001	▶*p* = 0.02	▶*p* = 0.003	▶*p* = 0.001
No	325(66.7)	26 (8.2)	17 (5.3)	32 (9.9)	36 (11.1)	7 (2.2)
Yes	162(33.3)	25 (15.8)	22 (13.9)	28 (17.4)	34 (21.0)	14 (8.6)
Spilling on body/ clothing		▶*p* < 0.001	▶*p* < 0.001	▶*p* = 0.047	▶*p* < 0.005	▶*p* < 0.001
No	389 (79.6)	31 (8.2)	22 (5.8)	42 (10.8)	47 (12.1)	8 (2.1)
Yes	100(20.4)	20 (20.8)	17 (17.7)	18 (18.2)	23 (23.0)	13 (13.0)
Forms of exposure		*p* = 0.009 *	*p* = 0.005 *	*p* = 0.02 *	*p* = 0.002 *	*p* < 0.001 +
Up to 4 forms	128 (28.3)	6 (4.8)	5 (4.0)	10 (7.9)	9 (7.0)	0
5–6 forms	140 (30.9)	16 (11.7)	9 (6.6)	17 (12.1)	20 (14.3)	4 (2.9)
7–12 forms	185 (40.8)	26 (14.4)	23 (12.7)	31 (16.8)	36 (19.5)	16 (8.6)
Multi-chemical exposure		*p* < 0.001 *	*p* < 0.001 *	▶*p* = 0.56	*p* = 0.01 *	*p* = 0.03 +
Used up to 2 types	143(29.1)	9 (6.5)	6 (4.3)	16 (11.2)	14 (9.8)	2 (1.4)
Used 3–4 types	189(38.4)	12 (6.5)	8 (4.3)	21 (11.1)	24 (12.7)	7 (3.7)
5 types or more	160(32.5)	30 (19.2)	25 (16.0)	23 (14.6)	32 (20.0)	12 (7.5)
Frequency of exposure/month		*p* = 0.006 *	*p* = 0.003 *	▶*p* = 0.86	*p* = 0.07 *	▶*p* = 0.23
Up to 2 days/month	212(43.8)	14 (6.8)	9 (4.3)	25 (11.8)	25 (11.8)	6 (2.8)
3–4 days/month	153(31.6)	18 (12.0)	14 (9.3)	18 (11.8)	22 (14.4)	10 (6.5)
5 days/month or more	119(24.6)	19 (16.4)	16 (13.8)	16 (13.7)	23 (19.3)	5 (4.2)
Always wear PPE		*p* = 0.01 *	*p* = 0.04 *	▶*p* = 0.90	▶*p* = 0.97	▶*p* = 0.69
No/Uses one PPE	122(24.8)	20 (17.2)	16 (13.8)	14 (11.5)	18 (14.8)	6 (4.9)
Two/three PPEs	209(45.5)	19 (9.3)	13 (6.3)	27 (13.0)	30 (14.4)	7 (3.3)
Four/five PPEs	161(37.2)	12 (7.6)	10 (6.3)	19 (11.8)	22 (13.7)	8 (5.0)

PRS = pesticide-related symptoms; BMI = body mass index; **▶** chi-square test; * linear trend test; + Fisher’s exact test. Note: unknown data were excluded.

**Table 2 ijerph-20-02818-t002:** Pesticide-related symptoms (PRS) *—comparison between period of low and high exposure to pesticides (*n* = 492).

PRS *	Period of Pesticide Exposure		RR (95%CI)
	Low ^#^	High ^&^	
	*n* (%)	*n* (%)	
Total	492 (100.0)	492 (100.0)	
Loss of appetite	5 (1.0)	18 (3.7)	3.60 (1.35–9.62)
Headache	10 (2.0)	35 (7.1)	3.50 (1.75–6.99)
Irritated eyes	10 (2.0)	33 (6.7)	3.29 (1.64–6.59)
Salivation	5 (1.0)	16 (3.3)	3.19 (1.18–8.65)
Dyspepsia/digestion difficult	6 (1.2)	17 (3.5)	2.82 (1.12–7.08)
Blurred vision	7 (1.4)	19 (3.9)	2.71 (1.15–6.40)
Skin irritation/allergy	8 (1.6)	19 (3.9)	2.38 (1.05–5.38)
Dizziness	11 (2.2)	23 (4.7)	2.10 (1.03–4.25)
Skin burn	1 (0.2)	5 (1.0)	5.00 (0.59–42.64)
Tremors	2 (0.4)	8 (1.6)	3.99 (0.85–18.70)
Vomiting	3 (0.6)	10 (2.0)	3.34 (0.93–12.06)
Sweating	4 (0.8)	11 (2.2)	2.76 (0.88–8.60)
Paresthesia	6 (1.2)	15 (3.1)	2.51 (0.98–6.40)
Intense weakness	6 (1.2)	15 (3.1)	2.51 (0.98–6.39)
Cramps	2 (0.4)	5 (1.0)	2.51 (0.49–12.88)
Phlegm	4 (0.8)	9 (1.8)	2.25 (0.70–7.27)
Cough	4 (0.8)	8 (1.6)	2.10 (0.61–6.63)
Nausea/feeling sick	6 (1.2)	11 (2.2)	1.83 (0.68–4.92)
Shortness of breath	6 (1.2)	11 (2.2)	1.83 (0.68–4.92)
Palpitation	8 (1.6)	12 (2.4)	1.50 (0.62–3.64)
Tearing	4 (0.8)	6 (1.2)	1.50 (0.43–5.28)
Restlessness/irritability	16 (3.3)	20 (4.1)	1.25 (0.65–2.37)
Wheezing	4 (0.8)	5 (1.0)	1.25 (0.34–4.63)
Abdominal pain	5 (1.0)	6 (1.2)	1.20 (0.37–3.90)
Diarrhea	0	1 (0.4)	§

Note: RR—risk ratio (95%CI = 95% confidence interval); * In the last 15 days, has felt onset of symptoms or worsening symptoms up to 48 h after using pesticides; # Low pesticide exposure—first stage of the field work; & High pesticide exposure—second stage of the field work. § Risk Ratio Undefined.

**Table 3 ijerph-20-02818-t003:** Prevalence of Pesticide-Related Symptoms (PRS) according to chemical types used in the last month.

PRS Chemical Group	Total *n* (%)	Frequent Headache *n* (%)	Irritated Eyes *n* (%)	Dizziness *n* (%)	Restless-ness/Irritability n (%)	Skin Allergy *n* (%)	Blurred Vision *n* (%)	Loss of Appetite *n* (%)	Dyspepsia *n* (%)	Salivation *n* (%)	Paresthesia *n* (%)	Intense Weakness *n* (%)	Palpitation *n* (%)	
Total sample	492 (100)	35 (7.1)	33 (6.7)	23 (4.7)	20 (4.1)	19 (3.9)	19 (3.9)	18 (3.7)	17 (3.5)	16 (3.3)	15 (3.1)	15 (3.1)	12 (2.4)	
Neonicotinoids	264 (53.7)	16 (6.1)	20 (7.6)	12 (4.6)	12 (4.5)	9 (3.4)	13 (4.9)	9 (3.4)	11 (4.2)	9 (3.4)	7 (2.7)	10 (3.8)	6 (2.3)	
Organophosphates	239 (48.6)	14 (5.9)	20 (8.4)	12 (5.0)	8 (3.3)	11 (4.6)	8 (3.3)	10 (4.2)	10 (4.2)	12 (5.0) *	4 (1.7) °+	8 (3.4)	7 (2.9)	
Pyrethroids	73 (14.8)	6 (8.2)	6 (8.2)	6 (8.3)	5 (6.8)	5 (6.8)	6 (8.2) *	3 (4.1) +	4 (5.5) +	6 (8.2) *	7 (9.6) ***	2 (2.7) +	3 (4.1) +	
Carbamates	55 (11.2)	3 (5.5) +	5 (9.1)	2 (3.6) +	3 (5.5) +	4 (7.3) +	4 (7.3) +	2 (3.6) +	1 (1.8) +	2 (3.6) +	3 (5.5) +	0 (0.0) +	3 (5.5) +	
Benzoylurea	55 (11.2)	3 (5.5) +	6 (10.9)	2 (3.6) +	3 (5.5) +	3 (5.5) +	3 (5.5) +	4 (7.3) +	3 (5.5) +	0 (0.0) +	0 (0.0) +	2 (3.6) +	1 (1.8) +	
Fipronil	42 (8.5)	3 (7.1) +	2 (4.8) +	1 (2.4) +	3 (7.1) +	2 (4.8) +	1 (2.4) +	3 (7.1) +	4 (9.5) *+	1 (2.4) +	2 (4.8) +	3 (7.1) +	1 (2.4) +	
Clomazone	278 (56.5)	19 (6.9)	18 (6.5)	9 (3.2) °	9 (3.2)	9 (3.2)	8 (2.9)	11 (4.0)	7 (2.5)	6 (2.2)	9 (3.2)	5 (1.8) °	6 (2.2)	
Glyphosate	150 (30.5)	14 (9.3)	10 (6.7)	10 (6.7)	8 (5.3)	5 (3.3)	6 (4.0)	(4.0) +	4 (2.7) +	9 (4.7)	6 (4.0)	6 (4.0)	4 (2.7) +	
Sulfentrazone	126 (25.6)	12 (9.5)	12 (9.5)	9 (7.2)	8 (6.3)	7 (5.6)	9 (7.1) *	7 (5.6)	7 (5.6)	6 (4.8)	7 (5.6) °	8 (6.3) **	3 (2.4) +	
Triazines	39 (7.9)	5 (12.8)	1 (2.6) +	0 (0.0) +	1 (2.6) +	1 (2.6) +	0 (0.0) +	0 (0.0) +	2 (5.1) +	1 (2.6) +	1 (2.6) +	0 (0.0) +	0 (0.0) +	
Dithiocarbamates	188 (38.2)	21 (11.2) **	20 (10.7) **	16 (8.5) **	14 (7.4) **	12 (6.4) *	11 (5.9) °	8 (4.3)	11 (5.9) *	11 (5.9) **	6 (3.2)	7 (3.7)	9 (4.8) **	
Metalaxyl	149 (30.3)	15 (10.1)	15 (10.1) *	10 (6.7)	9 (6.0)	6 (4.0)	8 (5.4)	4 (2.7) +	5 (3.4)	6 (4.0)	3 (2.0) +	5 (3.4)	6 (4.0)	
Iprodione	100 (20.3)	9 (9.0)	15 (15.0) ***	6 (6.0)	8 (8.0) *	4 (4.0) +	4 (4.0) +	3 (3.0) +	5 (5.0)	5 (5.0)	4 (4.0) +	3 (3.0) +	2 (2.0) +	
Copper compounds	26 (5.3)	3 (11.5) +	3 (12.0) +	2 (7.7) +	0 (0.0) +	0 (0.0) +	0 (0.0) +	0 (0.0) +	0 (0.0) +	2 (7.7) +	3 (11.5) *+	0 (0.0) +	0 (0.0) +	
PRS Chemical group	*n* (%)	Nausea/Feeling Sick *n* (%)	Sweating *n* (%)	Shortness of Breath *n* (%)	Vomiting *n* (%)	Phlegm *n* (%)	Tremors *n* (%)	Cough *n* (%)	Abdominal Pain *n* (%)	Tearing *n* (%)	Cramps *n* (%)	Wheezing *n* (%)	Skin Burn *n* (%)	Diarrhea *n* (%)
Total		11 (2.2)	11 (2.2)	11 (2.2)	10 (2.0)	9 (1.8)	8 (1.6)	8 (1.6)	6 (1.2)	6 (1.2)	5 (1.0)	5 (1.0)	5 (1.0)	4 (0.8)
Neonicotinoids	264 (53.7)	6 (2.3)	2 (2.7) +	6 (2.3)	6 (2.3)	6 (2.3)	5 (1.9)	3 (1.1) +	4 (1.5) +	4 (1.5) +	3 (1.1) +	4 (1.5) +	4 (1.5) +	2 (1.1) +
Organophosphates	239 (48.6)	6 (2.5)	7 (2.9)	5 (2.1)	5 (2.1)	5 (2.1)	4 (1.7) +	5 (2.1)	3 (1.3) +	3 (1.3) +	2 (0.8) +	2 (0.8) +	2 (0.8) +	3 (1.3) +
Pyrethroids	73 (14.8)	4 (5.5) °+	3 (4.1) +	4 (5.5) °+	4 (5.5) *+	3 (4.1) +	4 (5.5) *+	1 (1.4) +	2 (2.7) +	2 (2.7) +	1 (1.4) +	2 (2.7) +	3 (4.1) *+	1 (1.4) +
Carbamates	55 (11.2)	3 (5.5) +	3 (5.5) +	3 (5.5) +	1 (1.8)	3 (5.5) °+	0 (0.0) +	2 (3.6) +	0 (0.0) +	2 (3.6) +	1 (1.8) +	1 (1.8) +	1 (1.8) +	0 (0.0) +
Benzoylurea	55 (11.2)	1 (1.8) +	0 (0.0) +	0 (0.0) +	1 (1.8) +	3 (5.5) °+	0 (0.0) +	2 (3.6) +	0 (0.0) +	1 (1.8) +	0 (0.0) +	0 (0.0) +	2 (3.6) °+	1 (1.8) +
Fipronil	42 (8.5)	2 (4.8) +	1 (2.4) +	3 (7.1) °+	0 (0.0) +	0 (0.0) +	0 (0.0) +	0 (0.0) +	1 (2.4) +	0 (0.0) +	3 (7.1) **+	0 (0.0) +	1 (2.4) +	0 (0.0) +
Clomazone	278 (56.5)	6 (2.2)	5 (1.8)	4 (1.4) +	6 (2.2)	6 (2.2)	3 (1.1) +	5 (1.8)	3 (1.1) +	4 (1.4) +	2 (0.7) +	2 (0.7) +	4 (1.4) +	2 (0.7) +
Glyphosate	150 (30.5)	6 (4.0) °	3 (2.0) +	5 (3.3) +	4 (2.7) +	4 (2.7) +	3 (2.0) +	5 (3.4) *	1 (0.7) +	2 (1.3) +	2 (1.3) +	3 (2.0) +	2 (1.3) +	1 (0.7) +
Sulfentrazone	126 (25.6)	5 (4.0)	4 (3.2) +	4 (3.2) +	6 (4.8) **	5 (4.0) *	5 (4.0) *	4 (3.2) +	3 (2.4) +	5 (4.0) ***	3 (2.4) +	4 (3.2) *+	4 (3.2) *+	2 (1.6) +
Triazines	39 (7.9)	0 (0.0) +	1 (2.6) +	0 (0.0) +	0 (0.0) +	0 (0.0) +	0 (0.0) +	0 (0.0) +	0 (0.0) +	0 (0.0) +	0 (0.0) +	0 (0.0) +	0 (0.0) +	0 (0.0) +
Dithiocarbamates	182 (37.0)	6 (3.2)	7 (3.7) °	9 (4.8) **	5 (2.7)	5 (2.7)	4 (2.1) +	4 (2.1) +	3 (1.6) +	3 (1.6) +	3 (1.6) +	4 (2.1) °+	1 (0.5) +	3 (1.6) +
Metalaxyl	149 (30.3)	4 (2.7) +	3 (2.0) +	6 (4.0) °	3 (2.0) +	3 (2.0) +	1 (0.7) +	2 (1.3) +	1 (0.7) +	0 (0.0) +	1 (0.7) +	3 (2.0) +	1 (0.7) +	1 (0.7) +
Iprodione	100 (20.3)	4 (4.0) +	5 (5.0) *	2 (2.0) +	2 (2.0) +	1 (1.0) +	1 (1.0) +	0 (0.0) +	1 (1.0) +	1 (1.0) +	2 (2.0) +	1 (1.0) +	1 (1.0) +	1 (1.0) +
Copper compounds	26 (5.3)	0 (0.0) +	3 (11.5) *+	0 (0.0) +	1 (3.8) +	1 (3.8) +	0 (0.0) +	0 (0.0) +	2 (7.7) *+	1 (3.8) +	1 (4.0) +	0 (0.0) +	0 (0.0) +	1 (3.8) +

Note: ° *p* ≤ 0.10; * *p* ≤ 0.05; ** *p* ≤ 0.01; *** *p* ≤ 0.001 Chi-square test; + Fisher’s exact test; PRS—pesticide-related symptoms.

**Table 4 ijerph-20-02818-t004:** Association between socioeconomic factors and occupational exposure to pesticides with several criteria of acute pesticide poisoning—Crude analysis through Poisson Regression (*n* = 492).

Variables	2 or More PRS	3 or More PRS	Medical Diagnosis	Toxicologist Possible Case	Toxicologist Probable Case
Socioeconomic	Crude PR (CI)	Crude PR (CI)	Crude PR (CI)	Crude PR (CI)	Crude PR (CI)
Sex	▶*p* = 0.001	▶*p* = 0.002	▶*p* = 0.11	▶*p* = 0.05	▶*p* = 0.04
Male	1	1	1	1	1
Female	2.38 (1.42–3.98)	2.62 (1.45–4.76)	1.51 (0.92–2.50)	1.59 (0.99–1.17)	2.43 (1.05–5.63)
Tobacco production	*p* = 0.006 *	▶*p* = 0.07	*p* = 0.009 *	*p* = 0.08 *	▶*p* = 0.03
Up to 4000 kg	1	1	1	1	1
4001–9000 kg	0.94 (0.56–1.63)	1.29 (0.68–2.47)	0.63 (0.37–1.07)	0.80 (0.49–1.31)	0.29 (0.10–0.86)
Above 9000 kg	0.32 (0.13–0.76)	0.42 (0.15–1.13)	0.44 (0.23–0.85)	0.61 (0.34–1.08)	0.36 (0.12–1.08)
Alcohol consumption	*p* = 0.03 *	*p* = 0.02 *	*p* = 0.02 *	*p* = 0.08 *	*p* = 0.02 *
Does not drink	1	1	1	1	1
Up to 7 days/month	0.65 (0.37–1.15)	0.57 (0.30–1.11)	0.90 (0.53–1.53)	0.73 (0.45–1.20)	0.40 (0.16–0.99)
≥8 days/month	0.45 (0.22–0.94)	0.38 (0.16–0.90)	0.44 (0.21–0.93)	0.58 (0.32–1.06)	0.24 (0.07–0.85)
BMI	▶*p* = 0.58	▶*p* = 0.45	*p* = 0.13 *	▶*p* = 0.13	▶*p* = 0.88
Normal Weight	1	1	1	1	1
Overweight	1.34 (0.77–2.34)	1.45 (0.77–2.75)	1.36 (0.81–2.29)	1.55 (0.98–2.45)	1.12 (0.46–2.69)
Obese	1.26 (0.56–2.82)	0.95 (0.33–2.72)	1.60 (0.81–3.17)	0.96 (0.44–2.09)	0.77 (0.17–3.41)
Main Forms of Exposure to Pesticides					
Applying Pesticides	▶*p* = 0.02	▶*p* = 0.03	▶*p* = 0.10	▶*p* = 0.07	▶*p* = 0.06
No	1	1	1	1	1
Yes	2.14 (1.15–3.99)	1.98 (1.07–4.53)	1.55 (0.92–2.62)	1.56 (0.96–2.51)	2.83 (0.97–8.29)
Preparing mixture	▶*p* = 0.06	▶*p* = 0.17	▶*p* = 0.11	▶*p* = 0.06	▶*p* = 0.11
No	1	1	1	1	1
Yes	2.08 (0.96–4.49)	1.80 (0.77–4.21)	1.70 (0.89–3.26)	1.79 (0.97–3.29)	3.22 (0.76–13.62)
Washing contaminated clothing	▶*p* = 0.03	▶*p* = 0.03	▶*p* = 0.02	▶*p* = 0.14	▶*p* = 0.02
No	1	1	1	1	1
Yes	1.77 (1.05–3.00)	1.98 (1.07–3.68)	1.76 (1.08–2.87)	1.40 (0.90–2.16)	3.02 (1.19–7.65)
Reentering after application	▶*p* = 0.02	▶*p* = 0.01	▶*p* = 0.22	▶*p* = 0.78	▶*p* = 0.97
No	1	1	1	1	1
Yes	1.99 (1.14–3.50)	2.32 (1.18–4.55)	1.35 (0.83–2.20)	1.06 (0.69–1.64)	0.98 (0.43–2.28)
Clothes wet during application	▶*p* = 0.01	▶*p* = 0.002	▶*p* = 0.02	▶*p* = 0.004	▶*p* = 0.002
No	1	1	1	1	1
Yes	1.94 (1.16–3.24)	2.61 (1.42–4.76)	1.76 (1.10–2.82)	1.90 (1.23–2.91)	4.01 (1.65–9.75)
Spilling on body/ clothing	▶*p* < 0.001	▶*p* < 0.001	▶*p* = 0.05	▶*p* = 0.005	▶*p* < 0.001
No	1	1	1	1	1
Yes	2.55 (1.53–4.28)	3.05 (1.69–5.53)	1.68 (1.01–2.79)	1.90 (1.22–2.98)	6.32 (2.69–14.83)
Forms of exposure ^(1)^	*p* = 0.007 *	*p* = 0.008 *	*p* = 0.02 *	*p* = 0.002 *	
Up to 4 forms	1	1	1	1	(a)
5–6 forms	2.45 (0.99–6.07)	1.66 (0.57–4.81)	1.54 (0.73–3.24)	2.03 (0.96–4.30)	(a)
7 forms or more	3.02 (1.28–7.11)	3.20 (1.25–8.20)	2.14 (1.09–4.21)	2.77 (1.38–5.54)	(a)
Multi–chemical Exposure ^(2)^	*p* < 0.001 *	*p* < 0.001 *	▶*p* = 0.56	*p* = 0.01 *	*p* = 0.01 *
Used up to 2 types	1	1	1	1	1
Used 3–4 types	1.00 (0.43–2.29)	1.00 (0.35–2.80)	0.99 (0.54–1.83)	1.30 (0.70–2.42)	2.65 (0.56–12.56)
Used 5 types or more	2.95 (1.45–5.99)	3.69 (1.56–8.72)	1.30 (0.72–2.36)	2.04 (1.14–3.67)	5.36 (1.22–23.55)
Frequency of use	*p* = 0.006 *	*p* = 0.002 *	▶*p* = 0.86	*p* = 0.07 *	▶*p* = 0.25
Up to 2 days/month	1	1	1	1	1
3–5 days/month	1.77 (0.91–3.45)	2.15 (0.95–4.83)	1.00 (0.57–1.76)	1.22 (0.72–2.08)	2.31 (0.86–6.22)
>5 days/month	2.42 (1.26–4.67)	3.17 (1.45–6.95)	1.16 (0.65–2.08)	1.64 (0.98–2.76)	1.49 (0.46–4.76)
Always wear PPE	*p* = 0.001 *	▶*p* = 0.04	▶*p* = 0.90	▶*p* = 0.97	▶*p* = 0.69
No/One PPE	1	1	1	1	1
Two/three PPEs	0.54 (0.30–0.97)	0.46 (0.23–0.92)	1.14 (0.62–2.08)	0.97 (0.57–1.67)	0.68 (0.23–1.98)
Four/five PPEs	0.44 (0.22–0.87)	0.46 (0.22–0.97)	1.03 (0.54–1.97)	0.93 (0.52–1.65)	1.01 (0.36–2.84)

Note: ▶ *p*-value of the Wald heterogeneity test; * *p*-value of linear trend test; ^(1)^ Sum of the 12 forms of exposure; ^(2)^ Sum of the types of chemicals used; (a) not possible to perform regression (zero category); PRS—pesticide-related symptoms; PPE—personal protective equipment; PR—prevalence ratio; CI—95% confidence interval.

**Table 5 ijerph-20-02818-t005:** Association between socioeconomic factors and occupational exposure to pesticides with several criteria of acute pesticide poisoning—adjusted analysis through Poisson regression (*n* = 492).

Variables	2 or More PRS	3 or More PRS	Medical Diagnosis	Toxicologist Possible Case	Toxicologist Probable Case
Socioeconomic	Adjusted PR (CI)	Adjusted PR (CI)	Adjusted PR (CI)	Adjusted PR (CI)	Adjusted PR (CI)
Sex	▶*p* = 0.001	▶*p* = 0.002	▶*p* = 0.12	▶*p* = 0.04	▶*p* = 0.04
Male	1	1	1	1	1
Female	2.31 (1.39–3.85)	2.57 (1.43–4.61)	1.48 (0.91–2.42)	1.60 (1.02–2.51)	2.34 (1.03–5.29)
Tobacco production	*p* = 0.04 *	*p* = 0.08 *	*p* = 0.03 *	*p* = 0.25 *	*p* = 0.04 *
Up to 4000 kg	1	1	1	1	1
4001–9000 kg	0.95 (0.56–1.62)	1.30 (0.69–2.46)	0.64 (0.38–1.08)	0.81 (0.50–1.32)	0.29 (0.10–0.87)
Above 9000 kg	0.33 (0.14–0.79)	0.43 (0.16–1.17)	0.45 (0.23–0.86)	0.62 (0.35–1.10)	0.37 (0.13–1.11)
Alcohol consumption	▶*p* = 0.73	▶*p* = 0.69	▶*p* = 0.12	▶*p* = 0.60	▶*p* = 0.26
Does not drink	1	1	1	1	1
Up to 7 days/month	0.93 (0.49–1.79)	0.85 (0.38–1.89)	1.02 (0.55–1.88)	0.83 (0.47–1.46)	0.48 (0.16–1.46)
≥8 days/month	0.73 (0.32–1.68)	0.64 (0.23–1.79)	0.50 (0.22–1.15)	0.70 (0.35–1.40)	0.29 (0.06–1.39)
Main Forms of Exposure to Pesticides					
Applying pesticides	▶*p* = 0.02	▶*p* = 0.03	▶*p* = 0.17	▶*p* = 0.09	▶*p* = 0.06
No	1	1	1	1	1
Yes	2.11 (1.16–3.86)	2.20 (1.10–4.43)	1.45 (0.86–2.44)	1.51 (0.94–2.44)	2.80 (0.98–7.99)
Preparing mixture	▶*p* = 0.03	▶*p* = 0.08	▶*p* = 0.07	▶*p* = 0.03	▶*p* = 0.06
No	1	1	1	1	1
Yes	2.33 (1.10–4.93)	2.09 (0.91–4.83)	1.80 (0.95–3.44)	1.93 (1.06–3.52)	3.81 (0.95–15.31)
Washing contaminated clothing	▶*p* = 0.45	▶*p* = 0.33	▶*p* = 0.16	▶*p* = 0.74	▶*p* = 0.11
No	1	1	1	1	1
Yes	1.24 (0.71–2.19)	1.40 (0.71–2.73)	1.45 (0.87–2.42)	1.09 (0.65–1.82)	2.23 (0.84–5.89)
Reentering after application	▶*p* = 0.03	▶*p* = 0.02	▶*p* = 0.23	▶*p* = 0.75	▶*p* = 0.90
No	1	1	1	1	1
Yes	1.86 (1.07–3.22)	2.17 (1.13–4.19)	1.35 (0.83–2.18)	1.07 (0.70–1.66)	0.95 (0.41–2.18)
Clothes wet during application	▶*p* = 0.01	▶*p* = 0.002	▶*p* = 0.02	▶*p* = 0.002	▶*p* = 0.002
No	1	1	1	1	1
Yes	1.94 (1.16–3.25)	2.65 (1.44–4.89)	1.72 (1.08–2.75)	1.99 (1.29–3.07)	4.11 (1.69–10.04)
Spilling on body/clothing	▶*p* < 0.001	▶*p* < 0.001	▶*p* = 0.02	▶*p* = 0.002	▶*p* < 0.001
No	1	1	1	1	1
Yes	2.76 (1.67–4.54)	3.44 (1.96–6.01)	1.82 (1.11–2.96)	1.99 (1.28–3.09)	6.52 (2.96–14.36)
Forms of exposure ^(1)^	*p* = 0.007 *	*p* = 0.007 *	*p* = 0.03 *	*p* = 0.02 *	
Up to 4 forms	1	1	1	1	(a)
5–6 forms	2.17 (0.89–5.29)	1.47 (0.51–4.18)	1.37 (0.65–2.85)	1.80 (0.85–3.83)	(a)
7 forms or more	2.92 (1.26–6.78)	3.16 (1.24–8.01)	2.04 (1.03–4.03)	2.76 (1.36–5.58)	(a)
Multi–chemical exposure ^(2)^	*p* < 0.001 *	*p* < 0.001 *	▶*p* = 0.53	*p* = 0.005 *	*p* = 0.001 *
Used up to 2 types	1	1	1	1	1
Used 3–4 types	1.16 (0.50–2.67)	1.19 (0.43–3.35)	1.08 (0.57–2.02)	1.38 (0.73–2.61)	3.23 (0.71–14.72)
Used 5 types or more	3.68 (1.77–7.65)	4.85 (2.03–11.59)	1.38 (0.73–2.60)	2.29 (1.25–4.19)	6.83 (1.65–28.33)
Frequency of pesticide use	*p* = 0.003 *	*p* = 0.001 *	▶*p* = 0.78	*p* = 0.04 *	▶*p* = 0.20
Up to 2 days/month	1	1	1	1	1
3–5 days/month	1.95 (1.01–3.76)	2.40 (1.08–5.34)	1.05 (0.60–1.84)	1.33 (0.78–2.27)	2.39 (0.91–6.30)
>5 days/month	2.46 (1.31–4.62)	3.24 (1.51–6.94)	1.23 (0.68–2.21)	1.73 (1.02–2.93)	1.52 (0.47–4.89)
Always wear PPE	*p* = 0.01 *	*p* = 0.04 *	▶*p* = 0.94	▶*p* = 0.87	▶*p* = 0.45
No/One PPE	1	1	1	1	1
Two/three PPEs	0.51 (0.28–0.91)	0.46 (0.22–0.93)	1.04 (0.57–1.88)	0.92 (0.54–1.56)	0.55 (0.19–1.57)
Four/five PPEs	0.42 (0.22–0.81)	0.45 (0.22–0.93)	0.94 (0.50–1.78)	0.86 (0.48–1.53)	0.94 (0.34–2.56)

Note: ▶*p*-value of the Wald heterogeneity test; * *p*-value of linear trend test; ^(1)^ Sum of the 12 forms of exposure; ^(2)^ Sum of the types of chemicals used; (a) not possible to perform regression (zero category); PRS—pesticide-related symptoms; PPE—personal protective equipment; PR—prevalence ratio; CI—95% confidence interval.

**Table 6 ijerph-20-02818-t006:** Sensitivity, specificity, positive/negative predictive value and agreement of PRS and medical diagnosis criteria for acute pesticide poisoning when compared with toxicology assessment. (Possible case = 14.2% and probable case = 4.3%).

Criterion	Sensitivity	Specificity	PPV	NPV	Kappa
**Toxicology Assessment of Possible Case**					
Two or more PRS	46.3%	95.2%	60.8%	91.6%	0.22 *
Three or more PRS	35.8%	96.2%	61.5%	90.2%	0.39 *
Medical Diagnosis	72.9%	97.9%	85.0%	95.6%	0.75 *
**Toxicology Assessment of Probable Case**					
Two or more PRS	79.0%	92.2%	29.4%	99.1%	0.19 *
Three or more PRS	79.0%	94.8%	38.5%	99.1%	0.49 *
Medical Diagnosis	90.5%	91.3%	31.7%	99.5%	0.43 *

Note: * Kappa Test *p*-value < 0.001; PPV—positive predictive value; NPV—negative predictive value; PRS—pesticide-related symptoms.

## Data Availability

The data presented in this study are not publicly available due to ethical reasons.

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
