# Peer review of "Acute Pesticide Poisoning in Tobacco Farming, According to Different Criteria"

_ijerph, 2023, doi:10.3390/ijerph20042818_

Round 1

Reviewer 1 Report

The manuscript by Faria et al. evaluates the prevalence of acute poisoning among family farm workers and attempts to analyze some factors associated with poisoning. Although the study was well conducted and the data analysis and interpretations were adequate, some aspects should be improved in this manuscript.

Lines 51-54

The authors should revise this paragraph, as they are mixing studies that evaluate acute exposure and others that evaluate chronic exposure. As is well known, the effects of long-term pesticide exposure differ markedly from acute exposures. 

Lines 125-131

Several words marked in bold appear in the text.

Methodology and results

The methodology is not sufficiently clear. The authors state that they have initially considered three stages. However, the results do not reference the third stage (collection stage). Additionally, it is not completely clear whether the analysis has been carried out on the exposed workers or their relatives or on both (although at some point in the methodology, it is indicated that it is carried out on the relatives). I consider that this should be clarified.

Limitations

I suggest that the authors consider this a study in which the worker auto-reports symptoms, which could be a limitation of the study. 

Reviewer 2 Report

The article addresses the incidence of acute pesticide toxicity among farmers who used pesticides. The work is very interesting, but it needs to be re-structured, so I qualify it as a minor revision.

Detailed comments are below:

-        Citation should be carefully checked and correct according to journal guidelines in whole manuscript

Methods

-          Please divide the content into subsections to make it more readable

-          Please indicate how many people participated in the study/survey

Results

-          Please divide the content into subsections analogous to the "Materials and Methods" section.

Conclusion:

Please indicate what are the challenges regarding the issue discussed, indicate other problems that were not mentioned in the article.
